# Combining Mg–Zn–Ca Bulk Metallic Glass with a Mesoporous Silica Nanocomposite for Bone Tissue Engineering

**DOI:** 10.3390/pharmaceutics14051078

**Published:** 2022-05-17

**Authors:** Yun Shin Chu, Pei-Chun Wong, Jason Shian-Ching Jang, Chih-Hwa Chen, Si-Han Wu

**Affiliations:** 1Graduate Institute of Biomedical Materials and Tissue Engineering, Taipei Medical University, Taipei 11031, Taiwan; shinshinjoyce@gmail.com; 2Department of Orthopedics, Taipei Medical University Hospital, Taipei 11031, Taiwan; s0925135546@gmail.com; 3Department of Orthopedics, School of Medicine, College of Medicine, Taipei Medical University, Taipei 11031, Taiwan; 4Orthopedics Research Center, Taipei Medical University Hospital, Taipei 11031, Taiwan; 5Graduate Institute of Materials Science and Engineering, National Central University, Taoyuan 32001, Taiwan; jscjang@ncu.edu.tw; 6Department of Mechanical Engineering, National Central University, Taoyuan 32001, Taiwan; 7School of Biomedical Engineering, College of Biomedical Engineering, Taipei Medical University, Taipei 11031, Taiwan; chihhwache@tmu.edu.tw; 8Department of Orthopedics, Taipei Medical University—Shuang Ho Hospital, New Taipei 11031, Taiwan; 9School of Medicine, College of Medicine, Taipei Medical University, Taipei 11031, Taiwan; 10Research Center of Biomedical Device, Taipei Medical University, Taipei 11031, Taiwan; 11Graduate Institute of Nanomedicine and Medical Engineering, Taipei Medical University, Taipei 11031, Taiwan; 12International PhD Program in Biomedical Engineering, College of Biomedical Engineering, Taipei Medical University, Taipei 11031, Taiwan

**Keywords:** mesoporous silica nanoparticles, Mg–Zn–Ca bulk metallic glass, bone tissue engineering, osteogenic growth peptide, osteogenic differentiation, osteoinduction, osteoconduction, osseointegration

## Abstract

Mg–Zn–Ca bulk metallic glass (BMG) is a promising orthopedic fixation implant because of its biodegradable and biocompatible properties. Structural supporting bone implants with osteoinduction properties for effective bone regeneration have been highly desired in recent years. Osteogenic growth peptide (OGP) can increase the proliferation and differentiation of mesenchymal stem cells and enhance the mineralization of osteoblast cells. However, the short half-life and non-specificity to target areas limit applications of OGP. Mesoporous silica nanoparticles (MSNs) as nanocarriers possess excellent properties, such as easy surface modification, superior targeting efficiency, and high loading capacity of drugs or proteins. Accordingly, we propose a system of combining the OGP-containing MSNs with Mg–Zn–Ca BMG materials to promote bone regeneration. In this work, we conjugated cysteine-containing OGP (cgOGP, 16 a.a.) to interior walls of channels in MSNs and maintained the dispersity of MSNs via PEGylation. An in vitro study showed that metal ions released from Mg–Zn–Ca BMG promoted cell proliferation and migration and elevated alkaline phosphatase (ALP) activity and mineralization. On treating cells with both BMG ion-containing Minimum Essential Medium Eagle-alpha modification (α-MEM) and OGP-conjugated MSNs, enhanced focal adhesion turnover and promoted differentiation were observed. Hematological analyses showed the biocompatible nature of this BMG/nanocomposite system. In addition, in vivo micro-computed tomographic and histological observations revealed that our system stimulated osteogenesis and new bone formation around the implant site.

## 1. Introduction

Recently, much attention has been focused on Mg–Zn–Ca bulk metallic glass (BMG) as orthopedic fixation implants to remedy the drawbacks of Mg metal, including hydrogen evolution and tissue inflammation [1,2,3,4,5,6]. With a single-phase structure, Mg–Zn–Ca BMG provides superior mechanical properties [7] and an appropriate degradation rate (<0.5 mm/year). Specifically, the Mg ions released from Mg–Zn–Ca BMG can promote osteoblast proliferation and migration, enhancing bone mineralization and ingrowth [6,8,9]. High biodegradability and osteoconductive properties make Mg–Zn–Ca BMG more able to solve secondary surgery issues [6]. In addition to advances in bone implants, achieving osseointegration is the general goal. The current study focused on developing bone implants as structural supports with simultaneous induction of bone regeneration [10]. Surface functionalization of implants is a practical strategy to provide stability of bone–implant contact [11] and prevent implant-associated infections [12,13,14]. Although both of these features stimulate bone formation, they are still passive methods in bone tissue regeneration.

Bone repair/regeneration is a complex process involving the participation of various cell types and biological states [15]. Osteoinductive materials with capabilities of recruitment and differentiation of bone mesenchymal stem cells (BMSCs) to osteoblasts are set to acquire high-quality bone [10]. Growth factors and drugs, such as bone morphometric protein (BMP)-2, insulin-like growth factor (IGF)-1, and dexamethasone, have also been utilized for additional treatment [15]. The osteogenic growth peptide (OGP) is an endogenous tetradecapeptide in mammalian serum related to bone repair/regeneration [16,17]. The C-terminal pentapeptide “YGFGG” [18] is the minimal amino acid sequence that retains the biologically active part of OGP [19,20]. It stimulates the proliferation of osteoblastic cells through the mitogen-activated protein kinase (MAPK)/extracellular signal-regulated kinase (ERK) pathway [21,22], participates in differentiation to upregulate osteogenesis, and supports bone formation [16,17]. However, the short half-life, limited tissue residence, and non-specificity to target areas may restrict practical applications of OGP through injections [23,24]. In addition, high doses often lead to side effects such as inflammatory reactions, neoplasia, and ectopic bone formation [15]. Thus, establishing a highly effective drug delivery system to convey OGP is critical.

As a promising drug carrier, mesoporous silica nanoparticles (MSNs) provide advantages in the protection and controlled release of therapeutics. In addition, both particle sizes and interior walls of MSNs are adjustable, and multiple drugs or proteins can be loaded on demand for various conditions [25,26,27,28]. Several types of silica nanoparticles have been enrolled in clinical trials for therapeutic or diagnostic use [29]. This work aims to achieve osseointegration by combining the osteoconductive Mg_66_Zn_29_Ca_5_ BMG and OGP-containing MSNs (Figure 1a). We synthesized PEGylated MSNs with polyethylenimine (PEI) modification (designated as MSN@PEG/PEI), conjugating sulfhydryl-OGP to amine-containing MSN@PEG/PEI via heterobifunctional crosslinker OPSS-PEG-NHS to obtain MSN@PEG/PEI-OGP. Then, we investigated the effects of MSN@PEG/PEI-OGP on MC3T3-E1 cell migration and invasion ability, osteogenic capacity, and mineralization. The released OGP from MSN@PEG/PEI-OGP in vitro positively regulated osteogenic differentiation via the MAPK/ERK pathway. Additionally, the addition of an extraction medium, which was obtained from immersing Mg_66_Zn_29_Ca_5_ BMG in α-MEM, showed enhanced cell proliferation and migration ability. Finally, we applied the nanocomposite combining Mg_66_Zn_29_Ca_5_ BMG with MSN@PEG/PEI-OGP in rabbits and evaluated bone formation through micro-CT scanning and histological analyses.

It was the first implementation of combining MSN with Mg_66_Zn_29_Ca_5_ BMG for orthopedic application. However, the copper mold casting method used in this study has remained challenging in manufacturing BMG with large dimensions and complex geometries. With bottom-up manufacturing techniques, several 3D printing techniques such as selective laser melting have been successfully applied to remedy these shortcomings of as-cast BMGs [30]. Based on the current inputs and validation, physician-scientists could develop examples of promising fixators by shaping the BMG into interference screws and endowing hybrid MSNs with synergistic effects for cruciate ligament reconstruction.

## 2. Materials and Methods

### 2.1. Chemicals and Reagents

All chemicals and reagents were obtained from commercial suppliers and used without further purification. Cetyltrimethylammonium bromide (CTAB, 99%+), ammonium hydroxide (NH_4_OH, 28~30% solution in water), tetraethyl orthosilicate (TEOS, 98%), and 3-aminopropyltrimethoxysilane (APTMS, 95%) were obtained from ACROS Organics™ (Waltham, MA, USA). Rhodamine isothiocyanate isomer (RITC, 70%) was purchased from Sigma-Aldrich (St. Louis, MO, USA). Orthopyridyldisulfide-polyethyleneglycol-N-hydroxysuccinimide (OPSS-PEG-NHS, MW 200) was purchased from Biochempeg (Watertown, MA, USA). n-Decane (C_10_H_22_, 99%) was purchased from Alfa Aesar (Ward Hill, MA, USA). 2-[Methoxy(polyethyleneoxy)6-9propyl]trimethoxysilane, tech-90 (PEG-silane, M.W. 460–590 g/mol) and trimethoxysilylpropyl modified (polyethyleneimine) (PEI-silane, M.W. 1500–1800 g/mol, 50% in isopropanol) were acquired from Gelest (Morrisville, PA, USA). 2-Iminothiolane hydrochloride (Traut’s reagent, ≥98% (TLC), powder) was purchased from Sigma-Aldrich. Ultrapure deionized (DI) water was generated using a Millipore Milli-Q Plus system (Merck Group, Darmstadt, Germany).

### 2.2. Mg–Zn–Ca Bulk Metallic Glass Preparation and Composition Analysis

An induction melting technique manufactured Mg_66_Zn_29_Ca_5_ BMG under an argon atmosphere environment, as described by a previous study [31]. Typically, ingots of Mg_66_Zn_29_Ca_5_ BMG were first melted using highly pure Mg, Zn, and Ca (>99.9%) by induction melting with the temperature of 700 °C for 10 min. Then, the ingot was cut into pieces and re-melted in a quartz tube and injected into a water-cooled Cu mold under an argon atmosphere to form a 3 mm in diameter and 6 mm long rod. In addition, #2000 sandpaper was used to polish the end face to ensure the surface’s flatness and roughness. Energy-dispersive spectroscopy (EDS) (Inspect F50; FEI, Waltham, MA, USA) was used to confirm that the composition of samples matched our original design.

### 2.3. Compression Test

The compression strength of Mg_66_Zn_29_Ca_5_ BMG rods was tested on a material test system (Hung Ta, HT9102, Taipei, Taiwan) at a strain rate of 10^−4^ s^−1^. In this experiment, samples were 6 mm long and 3 mm in diameter.

### 2.4. Degradation Behavior

Degradation behaviors of Mg_66_Zn_29_Ca_5_ BMG rods 6 mm long and 3 mm in diameter were subjected to an immersion test in simulated body fluid (SBF)-Hank’s solution (8.0 g NaCl, 0.4 g KCl, 0.14 g CaCl_2_, 0.35 g NaHCO_3_, 0.1 g MgCl_2_·6H_2_O, 0.06 g MgSO_4_·7H_2_O, 0.06 g KH_2_PO_4_, and 0.06 g Na_2_HPO_4_·12H_2_O dissolved in 1 L DI water). After different immersion periods (0, 1, 2, 3, 4, 5, 6, 7, 8, 9, 10, 11, and 12 weeks), samples were removed from the solution and rinsed with DI water, and then dried at 60 °C in an oven. Weight loss of samples was measured on an electronic balance, and the change in the pH value of Hank’s solution was measured with a pH meter. All tests were conducted three times to ensure repeatability. For the in vitro study, an extraction medium was prepared using Mg_66_Zn_29_Ca_5_ BMG for immersion in Minimum Essential Medium Eagle-alpha modification (α-MEM, Gibco™, Waltham, MA, USA) at 4 °C for 30 days. Then, the impurities in the medium were removed by a 0.45 μm filter, and the final product was stored at 4 °C.

### 2.5. Synthesis of MSN@PEG/PEI-OGP

The synthesis of pore-expanded MSN@PEG/PEI based on the ammonia-catalyzed reaction was prepared by co-condensation according to the previous studies [25,27,28,32]. Typically, 0.386 g of CTAB was dissolved in 160 g of 0.22 M aqueous ammonia solution at 50 °C. Then, 1.2 mL of decane and 15 mL of ethanol were added to the aqueous ammonia solution under continuously stirring. After 12 h, 2.5 mL of pre-conjugated RITC-APTMS and 3.3 mL of ethanolic TEOS (5 mL of TEOS in 20 mL of 99.5% ethanol) were added sequentially to the reaction solution with vigorous stirring. The pre-conjugated RITC-APTMS was prepared by combining 8 mg RITC and 10 μL of APTMS in 5 mL 99.5% ethanol under continuous stirring and dark conditions. After 1 h, the mixture of PEG-silane and PEI-silane (550 μL of PEG-silane and 20 μL of PEI-silane in 2 mL 99.5% ethanol) was introduced to the solution. After another 30 min, stirring was stopped and the obtained colloidal solution was aged at 50 °C for 20 h. The obtained particle solution was hydrothermally treated at 80 °C overnight. Then, surfactants were extracted with a mixture of 339 μL of HCl (36.5–38%)/20 mL 99.5% of ethanol at 60 °C for 1 h, and the template-removed MSN@PEG/PEI was stored in 30 mL of 90% ethanol. To prepare MSN@PEG/PEI-OGP, 150 μL of OPSS-PEG-NHS (14.3 mg/mL) was added to 1 mL of MSN@PEG/PEI (10 mg/mL) in 5 mM Na_2_HPO_4_ (pH 8.4) and stirred for 2 h at room temperature (R.T.). At the same time, we prepared the cysteine-containing (cg) OGP (16 a.a., Kelowna International Scientific Inc., New Taipei City, Taiwan) and Traut’s reagent at a molar ratio of 1:20 in 1 mL of phosphate-buffered saline (PBS; pH 7.4) for 30 min at R.T. to allow the peptide to acquire sulfhydryl-modified molecules. Afterward, MSN@PEG/PEI-OPSS and OGP at a molar ratio of 5:1 in 2 mL of PBS (pH 7.4) were mixed and stirred overnight at 4 °C. The obtained products of MSN@PEG/PEI-OGP were collected by centrifugation and washed several times with water to remove any unreacted crosslinker and peptide. To analyze the peptide concentrations of MSN@PEG/PEI-OGP, we used Tris (2-carboxyethyl) phosphine (TCEP, Sigma Aldrich, St. Louis, MO, USA) as a reducing agent to cleave disulfide bonds. Briefly, 15 μL of the TCEP solution (18 mg/mL of TCEP) was added to 1 mL of MSN@PEG/PEI-OGP and stirred at R.T. Peptide concentrations were determined by the Bradford assay (Bio-Rad, Hercules, CA, USA).

### 2.6. Characterization

Transmission electron microscopy (TEM, JEM-1230, JEOL, Akishima, Tokyo, Japan) was operated at 100 kV to acquire the morphology of the nanoparticles. TEM samples were prepared by dropping dispersed MSN@PEG/PEI (99.5% ethanol) onto carbon-coated copper grids and drying them in air. Dynamic light scattering (DLS) measurements were utilized to detect the hydrodynamic size distribution and zeta potential of the N.P.s, and samples were measured at least three times on a Nano ZS90 laser particle analyzer (Malvern Instruments, Malvern, UK) in different solvents (DI water and PBS). N_2_ adsorption-desorption isotherms were recorded by Micrometrics ASAP 2020 (Norcross, GA, USA). Samples were first degassed at 110 °C for 16 h. According to the Brunauer–Emmet–Teller (BET) equation and the standard Barrett–Joyner–Halenda (BJH) method, specific surface areas and pore size distribution plots were obtained.

### 2.7. In Vitro Study

#### 2.7.1. Culture of Cell Lines

The murine calvarial preosteoblasts subclone 14 cell line (MC3T3-E1) was purchased from American Type Culture Collection (ATCC, Manassas, VA, USA) and cultured in α-MEM with 10% fetal bovine serum (FBS), 1% 100 U/mL penicillin, and 100 μg/mL streptomycin at 37 °C in a humidified incubator with 5% CO_2_. The medium was renewed every 2~3 days and passaged when cells had reached approximately 80% confluence. For osteogenic differentiation, osteogenic differentiation medium (ODM) comprised of 50 μg/mL ascorbic acid (Sigma Aldrich) and 10 mM of glycerophosphoric acid disodium salt hydrate (Biosynth Carbosynth, Ltd., Newbury, UK) in a standard culture medium was used to induce MC3T3-E1 differentiation.

#### 2.7.2. Cell Viability and Proliferation Assay

To measure the cytotoxic response of MSN@PEG/PEI, a cell counting kit (CCK)-8 reagent (Dojindo Molecular Technologies, Rockville, MD, USA) was performed by manufacturer’s instructions. Briefly, MC3T3-E1 cells (3 × 10^3^ cells/well) were seeded in a 96-well plate and incubated at 37 °C in a 5% CO_2_ atmosphere. After cells had attached, 0~500 μg/mL of nanoparticles was added to the cells and then incubated at 37 °C with a 5% CO_2_ atmosphere for 24 h. Next, the culture medium was removed; a fresh medium with 10% CCK-8 was subsequently added and incubated at standard conditions for 1 h. The absorbance was measured at 450 nm using a microplate reader (model 680, Bio-Rad, Hercules, CA, USA). An indirect contact method carried out a biocompatibility test of Mg_66_Zn_29_Ca_5_ BMG. Briefly, samples of Mg_66_Zn_29_Ca_5_ BMG were immersed in α-MEM for 30 days at 4 °C. Then, MC3T3-E1 cells were seeded (3 × 10^3^ cells/well) into a 96-well plate and incubated at 37 °C with a 5% CO_2_ atmosphere. After 24 h, the culture medium was replaced with different ratios of an extraction medium and cultured in an incubator at 37 °C with a 5% CO_2_ atmosphere for another 24 and 72 h. The CCK-8 assays were conducted the in the same manner as previously described.

#### 2.7.3. Cellular Uptake

The cell uptake efficiency of MSN@PEG/PEI-OGP was detected by FACSCalibur flow cytometry (B.D. Biosciences, Franklin Lakes, NJ, USA) and fluorescence microscopy. RITC was conjugated to nanoparticles as a marker to quantify cellular uptake. MC3T3-E1 cells were seeded (3 × 10^5^ cells/well) into 6-well plates. After cell attachment (24 h), the culture medium was replaced with the desired concentration of nanoparticles in a serum-containing medium for 16 h. Then, cells were washed with PBS, harvested by trypsin/EDTA, and resuspended in PBS. A flow cytometric analysis was carried out, and a minimum of 10^4^ cells was acquired.

#### 2.7.4. Cell Movement

Here, scratch migration/wound-healing assays (Culture-Insert 2 well, ibidi cells in focus, Gräfelfing, Germany) and Transwell invasion/chamber assays (Corning, Corning, NY, USA) were used to investigate the impact of an extraction medium and MSN@PEG/PEI-OGP on MC3T3-E1 cell motility.

#### Scratch Migration Assay

Two experimental designs were conducted to study cell motility: (1) immediately after treatment (IAT) and (2) delayed after treatment (DAT) of the conditioned media. For the IAT experimental setup, cells were suspended in serum-free αMEM (SF-αMEM) at a density of 8 × 10^5^ cells/mL, then 70 μL of cell suspension was applied to each side of the Culture-Insert 2 well. After cell attachment, the insert was gently removed with sterile tweezers, and the media was replaced with different conditioned SF-αMEM. An additional 16 h was performed before the assay. For the DAT experimental setup, cells were first seeded into 6-well plates at a density of 5 × 10^5^ cells/well, and then treated with various conditioned αMEM for 24 h. The subsequent steps were the same as the first experimental set. All images of migrating cells were captured with an optical microscope, and Image J software (version 1.53K, National Institutes of Health, Bethesda, MD, USA) was used for quantification. The migration rate was determined as follows: $\mathrm{Migration}\;\mathrm{rate}\;\left(\%\right)=(A_{0}-A_{n})/A_{0}\times 100\%$, where *A*_0_ is the beginning wound area, and *A_n_* is the wound area 16 h after removing the Culture-Insert 2 well (Ibidi) [33].

#### Transwell Invasion Assay

Here, we further introduced the invasion assay to analyze the vertical movement of cells through a filter, representing the extracellular matrix. According to the manufacturer’s instructions, a Transwell chamber assay was applied. Cells were suspended in a culture medium at a density of 7.5 × 10^4^ cells/mL, and 200 μL of cell suspension was seeded in the insert (upper chamber, pore size 8 μm), which was placed in a 24-well plate. Afterward, 700 μL of different concentrations of an extraction medium was added to the opposite side (lower chamber) under standard conditions. After 16 h, the culture medium was removed from the insert, treated with 4% (wt) paraformaldehyde (PFA) for 15 min, and stained with 0.5% crystal violet for 20 min at room temperature. Then, non-migrated cells were gently washed and scraped off with cotton swabs. Notably, the invasion assay was conducted to study the cell invasion after conditioned media treatment. Cells were treated with various conditions for 24 h in 6-well plates. Afterward, cells were harvested by trypsin/EDTA, suspended in an SF medium at a density of 1.2 × 10^5^ cells/mL, and 200 μL of the suspension was seeded in the insert (upper chamber, pore size 8 μm). To create the attractant gradient, 700 μL of complete medium was placed on the bottom well (lower chamber). Subsequent steps and analyses were the same as those described above. All images of migrating cells were captured by an optical microscope and analyzed by Image J software (version 1.53K, National Institutes of Health, Bethesda, MD, USA) for quantification.

#### 2.7.5. Western Blot Assay

To further understand the mechanism of the cell migration ability, cells were seeded (2 × 10^6^ cells) into 100 mm^2^ cell culture dishes. After cell attachment, the culture medium was replaced with a fresh medium containing the desired conditions for 24 h. Cells were lysed with RIPA buffer containing phosphatase (10×) and protease inhibitors (200×) for 2 h on ice. After sonication, samples were centrifuged at 4 °C for 20 min, and the protein concentration was determined with a BCA Protein Assay Kit (Thermo, Waltham, MA, USA). Equal amounts of protein (50 μg) were, respectively, loaded and separated on 8% and 10% sodium dodecyl sulfate-polyacrylamide gel electrophoresis (SDS-PAGE) gels and then transferred onto polyvinylidene difluoride (PVDF) membranes (Merck Millipore, Burlington, MA, USA). Membranes were blocked in 5% bovine serum albumin (BSA) for 1 h and incubated at 4 °C overnight with primary antibodies: anti-α-tubulin (1:2000, Dallas, TX, USA), anti-phospho-focal adhesion kinase (FAK) ((Tyr397), 1:1000, Cell Signaling Technology, Danvers, MA, USA), anti-phospho-paxillin ((Tyr118), 1:500, Cell Signaling Technology), and anti-phospho-ERK 42/44 (1:2000, Cell Signaling Technology). After washing with TBST (1× Tris-buffered saline (TBS) and 0.1% (*v*/*v*) Tween 20), membranes were incubated with secondary antibodies for 1 h. Finally, bands were detected with an enhanced chemiluminescence (ECL) substrate kit (Amersham Pharmacia Biotech, G.E. Healthcare, Bucks, UK) and analyzed with the BioSpectrum AC imaging system (UPV, Analytik Jena AJ, Jena, Germany).

#### 2.7.6. Alkaline Phosphatase (ALP) Staining and Alizarin Red s (ARS) Staining

To analyze the effects of Mg_66_Zn_29_Ca_5_ BMG, MSN@PEG/PEI, and OGP on the early stage of osteogenic differentiation, ALP activity was determined by ALP staining (Sigma Aldrich). Briefly, cells were seeded at a density of 3 × 10^4^ cells/well. After cell attachment, the culture medium was replaced with a differentiation medium containing desired conditions. After culturing for 14 days, cells were washed with PBS and fixed with 4% PFA for 2 min. The fixative solution was aspirated, and cells were subsequently washed with water. ALP staining was carried out for 30 min, and images of cells were captured with an optical microscope, while Image J software was used for quantification. To analyze the effects of conditioned treatment on calcification deposition of MC3T3-E1 cells, cells were seeded at a density of 2 × 10^4^ cells/well. After cell attachment, the culture medium was replaced with a differentiation medium containing the desired condition. After being cultured for another 21 days, cells were washed with PBS and fixed with 4% PFA for 15 min. The fixative solution was aspirated, and cells were subsequently washed with water. ARS (1%, Sigma Aldrich) staining was carried out for 60 min, and images of cells were captured with an optical microscope, while Image J software was used for quantification. For quantitation, cells stained with ARS were dissolved in 10% cetylpyridinium chloride (Sigma Aldrich) for 1  h. The absorbance was measured at 562 nm with a microplate reader (model 680, Bio-Rad).

### 2.8. In Vivo Study

The animal experiment was carried out according to a protocol approved by the Institutional Animal Care and Use Committee (IACUC) of Taipei Medical University (approval no. LAC-2018-0164). In total, 12 male New Zealand white rabbits with a mean body weight of 3 kg were used in this study, and they were evaluated at 4 weeks post-surgery. The animals were randomly divided into Mg_66_Zn_29_Ca_5_ BMG with PBS and Mg_66_Zn_29_Ca_5_ BMG with MSN@PEG/PEI-OGP. There were three animals in each group.

#### 2.8.1. Surgical Method in Rabbits

Animal surgery was performed under isoflurane general anesthesia. The surgery was a lateral parapatellar arthrotomy on the condyles of both knee joints. A tunnel with a diameter of 2 mm and a length of 4 mm, perpendicular to the femur’s long axis, was created with an electric bone drill. Mg_66_Zn_29_Ca_5_ BMG rods were directly inserted into the tunnel, and then 50 μL of different conditions was injected into the tunnel (MSN@PEG/PEI-OGP: 10 mg/mL; OGP: 0.6 mg/mL). After surgery, the wound was sutured in layers to decrease the risk of an inflammatory reaction. Finally, an antibiotic (Enrofloxacin; 12 mg/kg; Bayer AG, Leverkusen, Germany) and an analgesic (Ketoprofen; 12 mg/kg; Nang-Kuang Pharmaceutical, Taipei, Taiwan) were given for 3 days.

#### 2.8.2. Blood Collection and Hematology

At 4 weeks post-surgery, blood samples were collected before the rabbits were sacrificed through exsanguination. Blood samples were collected in EDTA vacutainer tubes to analyze the complete blood count (CBC). A hematological study was conducted with a Hematology Analyzer (Procyte Dx; Idexx, Westbrook, ME, USA). The items measured included lymphocytes (LYMs), white blood cells (WBCs), neutrophils (NEUs), monocytes (MONOs), eosinophils (E.O.s), basophils (BASOs), red blood cells (RBCs), the mean corpuscular volume (MCV), and platelet count (PLT). Serum biochemistry was determined with a chemistry analyzer (VetTest; Idexx), and blood samples were centrifuged to collect serum and were stored at −20 °C until used. The items measured included creatinine (CREA), blood urea nitrogen (BUN), total bilirubin (TBIL), albumin (ALB), alanine aminotransferase (ALT), alkaline phosphatase (ALP), and magnesium (Mg).

#### 2.8.3. Micro-CT Scan and 3D Image Reconstruction

After being sacrificed, specimens were harvested from rabbits and scanned with micro-CT (SkyScan 1176, Bruker, Billerica, MA, USA). Micro-CT images were reconstructed with NRecon Reconstruction. The bone density was calculated using reconstruction data.

#### 2.8.4. Histological Observations

After micro-computed tomographic (CT) scanning, specimens from rabbits were fixed in formalin and subjected to decalcification treatment. Afterward, samples were rinsed in DI water and dehydrated with an ethanol series in the following order: 70%, 80%, 95%, and 99.9%. Then, specimens were cleaned with xylene twice and immersed in paraffin. After 2 h, specimens were paraffin embedded. Finally, sections were stained with hematoxylin and eosin (H&E) and Masson’s trichrome to observe the bone morphology.

### 2.9. Statistical Analysis

All statistical analyses were performed using GraphPad Prism 8 software (GraphPad, San Diego, CA, USA). Data are expressed as the mean ± standard deviation (S.D.), and three time-independent experiments were carried out. Student’s *t*-test evaluated statistical differences between the two groups, and a *p*-value of < 0.05 was considered statistically significant.

## 3. Results and Discussion

### 3.1. Characterization of Mg_66_Zn_29_Ca_5_ BMG

#### 3.1.1. Chemical Composition and Compressive Strength of Mg_66_Zn_29_Ca_5_ BMG

As shown in Table 1, the chemical composition of Mg_66_Zn_29_Ca_5_ BMG analyzed by EDS conforms with our design. There was no significant loss of raw materials during bulk metallic glass preparation. Additionally, Figure 1b shows that the compressive stress of Mg_66_Zn_29_Ca_5_ BMG was 680 MPa, consistent with our previous studies [1,31]. Here, copper mold casting, among the most effective procedures, was used to prepare corrosion-resistant BMG. However, the composition control, alongside the ability to produce customized implants, has proved challenging [34]. As a new generation technology for the development of BMG [35], additive manufacturing provides a promising, economical and scalable technique. Efforts have been ongoing to customize mass production without compromising the amorphous structure [36].

#### 3.1.2. Degradation Behavior of Mg_66_Zn_29_Ca_5_ BMG

Figure 1c shows the changes in pH value and the weight of Mg_66_Zn_29_Ca_5_ BMG while immersing in Hank’s solution within 6 weeks. A rapid pH increase, from pH 7.4 to pH 10.6, occurred during the first week of immersion. Then, the pH values saturated at approximately 11.0. After 6 weeks of immersion, the weight loss of Mg_66_Zn_29_Ca_5_ BMG reached 13.9% compared to their initial weight. The degradation rate as a function of different immersion times of Mg_66_Zn_29_Ca_5_ BMG is similar to our previous study [1]. For the in vitro study, the extraction medium was prepared by immersing Mg_66_Zn_29_Ca_5_ BMG in α-MEM at 4 °C for 30 days. Table 1 reveals that the extracted MEM, respectively, contained 60.26, 8.08, and 7.05 mg/L of Mg, Zn, and Ca ions. The elevated metal ion concentration confirmed that Mg_66_Zn_29_Ca_5_ BMG degraded over time. Here, magnesium is dominant among the released ions; however, the ratio among released metal ions differs from the composition of Mg_66_Zn_29_Ca_5_ BMG, especially the ratio between Mg and Zn. The result indicates that the degradation rate depends strongly on the microstructure and the microenvironment of Mg_66_Zn_29_Ca_5_ BMG.

### 3.2. Characterization of MSN@PEG/PEI-OGP

To obtain well-dispersed OGP-containing MSNs, we first introduced PEG and PEI on the surface of MSNs. Figure 2a TEM shows that MSN@PEG/PEI had an average diameter of 51.48 ± 9.59 nm (Figure 2b) and a porous hexagonal arrangement. DLS analysis (Figure 2c, Appendix A) revealed their hydrodynamic diameter in PBS was approximately 91.6 ± 0.7 nm. As shown in Figure 2d, the zeta potential versus pH titration curve is characteristic of MSNs with weak basic functional groups of PEI and exhibits an IEP at pH 8. To further determine the pore size and surface area of MSN@PEG/PEI, we performed N_2_ adsorption-desorption isotherm measurements. The BJH pore diameter and BET surface area of MSN@PEG/PEI were 3.38 nm and 593 m^2^/g, respectively (Figure 2e,f). Together with our previous study [27], the pore size of MSN@PEG/PEI is in between conventional MSNs (2.5 nm) and pore-expanded MSNs (3.9 nm), suggesting that interior surface modification of PEI was successfully performed. Next, to immobilize OGP on MSNs, we introduced a crosslinker OPSS-PEG-NHS to react with amine groups on MSN@PEG/PEI. OGP was covalently immobilized with the OPSS-end of MSN@PEG/PEI via a thiol/disulfide exchange reaction. The DLS measurement indicated that MSN@PEG/PEI-OGP suspended well in PBS solution, showing a slight increase in hydrodynamic diameter (107.8 ± 1.0 nm) compared to MSN@PEG/PEI (Figure 2c, Appendix A). The amount of OGP functionalized onto 1 mg of MSN@PEG/PEI-OGP determined by a spectrometric method was 60.9 μg. The peptide conjugation efficiency was calculated to be 47.3%.

### 3.3. Cell Viability and Cellular Uptake

To evaluate the in vitro biocompatibility of Mg_66_Zn_29_Ca_5_ BMG and MSN@PEG/PEI, MC3T3-E1 cells were cultured in different conditioned media for 24 h and then analyzed with CCK-8 assays. As shown in Figure 1d and Figure 2g, both the MSN@PEG/PEI and extracted-αMEM groups displayed comparable cell viability to the control group. Notably, on the 3rd day in Figure 1d, the viability rate was higher than that of the control group, starting with the group of 20% extracted-αMEM, and the trend rose with higher ratios of extracted-αMEM. Table 1 revealed that the extracted-αMEM was rich in Mg, Zn, and Ca ions, essential elements for cell growth in living systems [37]. Accordingly, the degradation of Mg_66_Zn_29_Ca_5_ BMG could promote cell proliferation. On the other hand, information on the size and internal density or granularity of the illuminated cells was acquired from flow cytometry to investigate the cellular uptake of fluorescent MSNs [38]. As demonstrated in Appendix A, regardless of whether OGP was conjugated to MSN@PEG/PEI, highly efficient cell uptake was observed due to the electrostatic interactions between the positive charged MSN@PEG/PEI and negative MC3T3-E1 cell membranes [32,39]. In addition, fluorescence images (Figure 2h) showed nanoparticles were mainly distributed in the cytoplasm.

### 3.4. In Vitro Migration Ability

It is known that the recruitment of BMSCs to implant sites and their derivation into osteoblasts are essential in bone healing [40]. Here, we first examined in vitro how extracted-αMEM, OGP, MSN@PEG/PEI, and MSN@PEG/PEI-OGP affect the cell movement of MC3T3-E1. Cell migration/invasion assays were performed in IAT and DAT experimental setups (Figure 3a). As shown in Figure 3b, cells gradually migrated to the scratch area over time in the IAT setup. Compared to normal αMEM, extracted-αMEM facilitated MC3T3-E1 migration in a concentration-dependent manner (Appendix A and Figure 2b). In the 20% extracted-αMEM (20% E-M) group, MC3T3-E1 cells mediated about a 1.5-fold increase in migration (Figure 3c). Additionally, Appendix A showed gradients of extracted-αMEM directed vertical cell migration through the Transwell membrane. The high magnesium concentrations in the extracted-αMEM surrounding cells promoted migration and proliferation [41,42,43]. On the other hand, the addition of OGP in αMEM also showed elevated cell motility. However, a decreased cell motility was observed in MSN@PEG/PEI group. The mechanism may be closely associated with regulating the phosphorylation of FAK (focal adhesion kinase), ERK (extracellular signal-regulated protein kinase), and paxillin, which are related to focal adhesion turnover and varies in vivo and in vitro [44]. Nevertheless, the quantitative analysis (Figure 3c) reveals that MSN@PEG/PEI-OGP rescued the decrease in migration and recovered cell motility to the original level, indicating that the released OGP could compensate for the hindrance of cell migration by MSN@PEG/PEI. Next, we co-treated cells with extracted-αMEM and MSN@PEG/PEI-OGP to mimic the degradation of Mg_66_Zn_29_Ca_5_ BMG in a clinical therapeutic setting. Notably, when cells were subjected to the co-treatment of Mg_66_Zn_29_Ca_5_ BMG and MSN@PEG/PEI-OGP, cell migration was significantly elevated and even higher than in the free OGP group. Alternatively, we study the cell motility in the DAT setup. Likewise, the most pronounced effect is visible in the co-treatment group in the Transwell invasion assay (Figure 3d,e). Retardation in cell movement also occurred in MSN@PEG/PEI group (Figure 3d,e and Appendix A). However, the quantitative analyses revealed that the cell migration area in the MSN@PEG/PEI-OGP group is higher than in the control group. In particular, all OGP-containing groups displayed a more robust enhancement in vertical than horizontal migration of cells (Figure 3c,e). Still, a more detailed study is needed to better understand the nature of OGP.

Next, we further investigated the molecular mechanisms of cell adhesion signaling in the IAT setup. In cell migration, successive adhesion turnover causes cells to protrude and move forward [45]. Numerous studies have reported that focal adhesion (F.A.) is required in cell migration, adhesion, spread, and reorganization of the actin cytoskeleton [45,46,47,48]. Among them, F.A. kinase (FAK) is a well-known mediator that regulates intracellular signaling and is involved in adhesion turnover. Extracellular-regulated protein kinases (ERKs) were suggested to be related to cell motility [47,48,49]. As shown in Figure 3f, expressions of phospho-Tyr 397, phospho-Tyr 118, and phospho-ERK 42/44 decreased in MC3T3-E1 after MSN@PEG/PEI treatment compared to the control group. In contrast, all levels increased across the groups containing extracted-αMEM and OGP. It has been known that phospho-Tyr 397 can recruit Src to compose the FAK-Src complex, which can phosphorylate paxillin at Tyr118 [48]. This, in turn, leads to the recruitment of inactive ERK, allowing focal adhesion turnover. On the contrary, once inactive ERK is not activated at the proper position, paxillin fails to bind to FAK, resulting in defective disassembly [48,50]. Together with the previous research, mechanisms involved in diminished migration and invasion of MSNs include enhancement of focal adhesion, induction of actin polymerization, and microtubule stabilization [46,51,52]. Here, we have not attempted to conduct an all-inclusive study. Accordingly, we introduced OGP on MSNs and focused on integrating this hybrid nanocarrier with bridgeable Mg_66_Zn_29_Ca_5_ BMG. OGP is known for ERK1/2 activation through the Gi protein receptor [21]. Figure 3 and Appendix A show that OGP released from MSN@PEG/PEI-OGP through cleaved disulfide bonds by cytosolic reductants (e.g., GSH) remarkably enhanced cell motility [53].

### 3.5. In Vitro Osteogenesis Ability

Among osteogenic differentiation markers, upregulation of ALP activity is considered vital in the early stage, while mineral deposition is regarded as the final stage of osteogenesis. We first studied the effect of extracted-αMEM and MSNs on the early stage of osteogenesis. ALP activity displayed in dark-purple color was performed after cells were cultured in different conditions for 14 days. Figure 4a,b show that a minor ALP activity occurred in the MSN@PEG/PEI group, while OGP and extracted-αMEM groups positively influenced osteoblast differentiation. Subsequently, calcium deposition was carried out on day 21 after conditioned media treatment. Figure 4c shows the extent of MC3T3-E1 calcification in different groups as visualized by ARS staining. The negative effect of MSN@PEG/PEI was observed. Either OGP or extracted-αMEM induced notable effects on cell calcification, whereas co-treatment of extracted-αMEM and MSN@PEG/PEI-OGP remarkably enhanced the calcification of MC3T3-E1. A quantitative comparison of various groups for calcium deposition was reported in Figure 4d, where the trend slightly differed in migration and invasion assays instead (Figure 3e). MSN@PEG/PEI-OGP remedied the inadequacy of MSN@PEG/PEI and performed better than 20% extracted-αMEM in osteogenic differentiation. Notably, the value of MSN@PEG/PEI-OGP was similar to that of free OGP, suggesting that MSNs with PEG modification could prevent OGP from hydrolysis and extend its half-life [54]. In addition, a large concentration gradient of GSH between the intracellular and the extracellular environment can trigger the release of OGP from MSN@PEG/PEI-OGP in cells, stimulating differentiation and mineralization [53].

### 3.6. In Vivo Biocompatibility

To evaluate the biocompatibility of Mg_66_Zn_29_Ca_5_ BMG and MSNs in vivo, we performed routine hematological tests, such as complete blood count and serum chemistry at 4 weeks post-surgery. As to CBC values (Appendix A), critical indicators of inflammatory reaction, including WBCs, NEUs, and MONOs were comparable to reference values. At the same time, Appendix A reveals similar biochemical parameters among all groups. Compared with the healthy rabbits, no apparent changes in creatinine levels, blood urea nitrogen, total bilirubin, albumin, alanine aminotransferase, and alkaline phosphatase for the rabbits with implant treatment of Mg_66_Zn_29_Ca_5_ BMG and co-treatment with either OGP or MSN@PEG/PEI-OGP. Notably, serum magnesium in all groups is in the normal range. Through the blood and body fluids, the main Mg ion component released from degradation could be stored in bones, soft tissues, serum, different types of cells and excreted via the urine [7]. Altogether, neither severe inflammatory reaction nor burdens on the kidneys and liver were observed after implantation of Mg_66_Zn_29_Ca_5_ BMG and MSNSs for 4 weeks.

### 3.7. In Vivo Bone Formation Ability

Previous studies have demonstrated that degraded magnesium alloys can increase bone mineral density (BMD) [6,7,55]; OGP can promote ALP activity and facilitate mineralization during osteogenesis in bone remolding [17,56]. Here, micro-CT scanning was applied to investigate BMD around the implantation site, and imaging reconstruction processes were conducted after treatment for 4 weeks. In Figure 4e, the bright white color represents the high-density regions of the Mg_66_Zn_29_Ca_5_ BMG implant, and the blue mark indicates newly formed bone. At 4 weeks post-surgery, little degradation of the implant was observed in all groups, while new bone formation attached around the implants was notable in MSN@PEG/PEI-OGP group. As shown in Figure 4f, the quantitative analysis confirmed that the combination of Mg_66_Zn_29_Ca_5_ BMG with MSN@PEG/PEI-OGP displayed the highest BMD value. Moreover, there is no difference between groups with and without the introduction of free OGP in BMD around the implantation site. The limited in vivo effect of OGP may be attributable to its nature of being quickly metabolized and short lasting in target areas. On the other side, benefiting from the properties of protecting drugs and being engulfed promptly by cells, MSN@PEG/PEI-OGP is suitable for durable and intracellular delivery of OGP.

### 3.8. In Vivo Histological Observations

To evaluate osteoinduction and osseointegration between the interface of bone tissue and the implants, H&E staining and Masson’s trichrome staining were applied to decalcified sections at 4 weeks post-surgery. In Figure 4g, H&E staining shows that few osteoblasts (ob) and osteocytes (oc) around the Mg_66_Zn_29_Ca_5_ BMG, suggesting the implant as an osteoconduction can recruit cells [6]. In the group of Mg_66_Zn_29_Ca_5_ BMG, few ob/oc cells were gathered in one corner of the implant, and bone formation was incomplete. Instead, numerous layers of osteoblasts evenly gathered around the implant were observed in the group of Mg_66_Zn_29_Ca_5_ BMG+MSN@PEG/PEI-OGP. As for Masson’s trichrome staining, the blue surrounding the implanted tissue represents collagenous tissue, regenerated bone, or osteoid, while the red indicates the mature bone. In Figure 4g, all groups showed little surface degradation of Mg_66_Zn_29_Ca_5_ BMG, enabling calcified bone deposited around the implant site and trending to ingrowth toward the center of the implant. Mg_66_Zn_29_Ca_5_ BMG with the addition of MSN@PEG/PEI-OGP displayed the highest number of mature bones encircled the bone–implant interface, indicating the intracellular delivery of OGP via MSN@PEG/PEI-OGP enhanced the bone regeneration. In contrast, much rougher interfacial morphology and less bone matrix were visible in the Mg_66_Zn_29_Ca_5_ BMG alone group, indicating limited interaction between the implant and surrounding tissue [57]. Collectively, Mg_66_Zn_29_Ca_5_ BMG as an osteoconductive implant enhanced osteoblasts recruitment, and with further incorporation of MSN@PEG/PEI-OGP facilitated osteogenesis in bone remodeling [10].

## 4. Conclusions

We demonstrated a proof-of-concept approach to leverage bone formation by combining bone implants with nanotherapeutics. Mg_66_Zn_29_Ca_5_ BMG possesses superior compressive strength as a source for bone implants. Metal ions released from degradation of Mg_66_Zn_29_Ca_5_ BMG increased ALP activity and calcium deposition in MC3T3-E1 cells. In addition, MSN@PEG/PEI-OGP exhibited efficient cellular uptake, promoting cell migration and osteogenesis. Moreover, in vivo micro-CT and histological observations revealed that MSN@PEG/PEI-OGP could stimulate osteogenesis and new bone formation around the implant site. Taken together, an initial attempt based on the integration of nanomaterials and implants was made to amplify the osteoconductive and osteoinductive effects to achieve the strong osseointegration ability for promising bone tissue engineering therapies. Coating MSN@PEG/PEI-OGP could achieve a practical application onto the Mg_66_Zn_29_Ca_5_ BMG interference screw to meet clinical research demands.

## Figures and Tables

**Figure 1 pharmaceutics-14-01078-f001:**
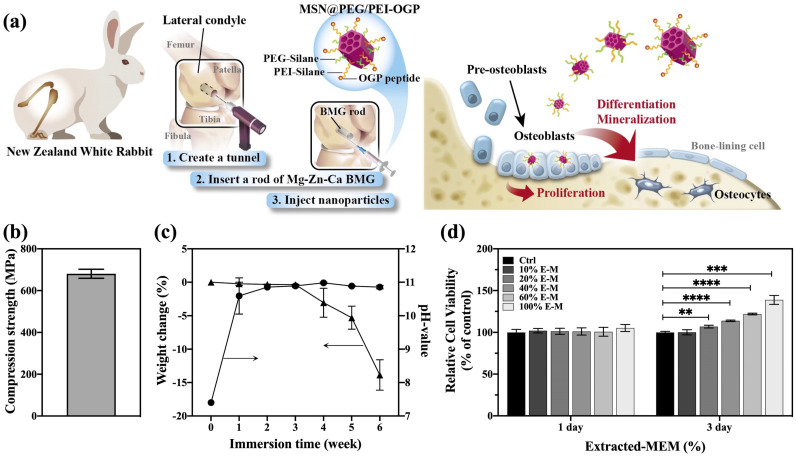
(**a**) Schematic illustration of Mg−Zn−Ca bulk metallic glass combined with mesoporous silica nanoparticle@PEG/PEI-osteogenic growth peptide for promoting bone regeneration in the tunnel. (**b**) Compressive strength measurement of Mg_66_Zn_29_Ca_5_ BMG. (**c**) Time−dependent degradation behavior of Mg_66_Zn_29_Ca_5_ BMG immersed in Hank’s solution. (**d**) Cell viability of MC3T3−E1 treated with different concentrations of extracted−αMEM. (*n* = 3 per group; data are the mean ± standard deviation, ** *p* < 0.01, *** *p* < 0.001, **** *p* < 0.0001). The circle represents the pH-value (right y-axis), and the triangle refers to the weight change (left y-axis) determined at each corresponding immersion time (x-axis).

**Figure 2 pharmaceutics-14-01078-f002:**
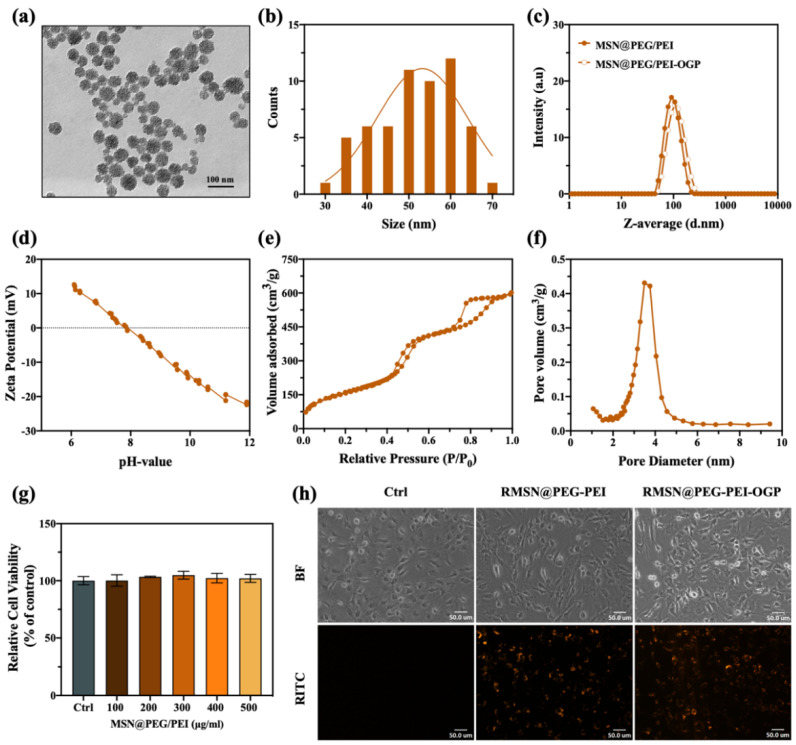
(**a**) TEM image and (**b**) particle size distributions of MSN@PEG/PEI. (**c**) DLS analyses of MSN@PEG/PEI and MSN@PEG/PEI−OGP in PBS. (**d**) Zeta potential measurements of MSN@PEG/PEI over pH ranging from 6 to 12. (**e**) Nitrogen adsorption-desorption isotherms and (**f**) pore size distribution of MSN@PEG/PEI. (**g**) Cell viability of MC3T3−E1 treated with different concentrations of MSN@PEG/PEI. (**h**) Fluorescence images of MC3T3−E1 cells at 24 h post−treatment.

**Figure 3 pharmaceutics-14-01078-f003:**
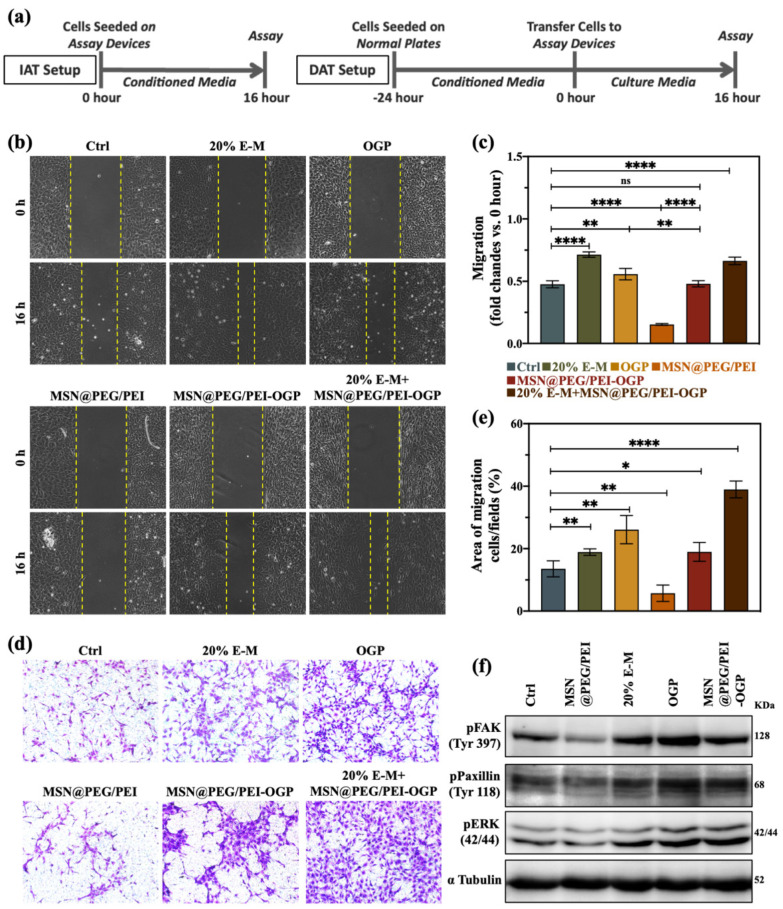
(**a**) Experimental design for cell migration/invasion study. (**b**) Scratch migration (wound healing) assay and (**c**) quantitative evaluation of MC3T3−E1 cells conducted immediately after treatment (IAT) of the conditioned media. (**d**) Transwell invasion assay and (**e**) quantitative evaluation of MC3T3−E1 cells conducted delayed after treatment (DAT) after treatment of the conditioned media. (**f**) Western blotting of phosphorylated (p)−focal adhesion kinase (FAK) (Tyr397), p−paxillin (Tyr118), p−extracellular signal−regulated kinase (ERK) (42/44), and α−tubulin expressions in MC3T3−E1 cells at 24 h post−treatment. The amount of nanoparticles was 100 μg/mL, and that of OGP was 6.06 μg/mL. (*n* = 3 per group). (Data are the mean ± standard deviation, * *p* < 0.05, ** *p* < 0.01, **** *p* < 0.0001).

**Figure 4 pharmaceutics-14-01078-f004:**
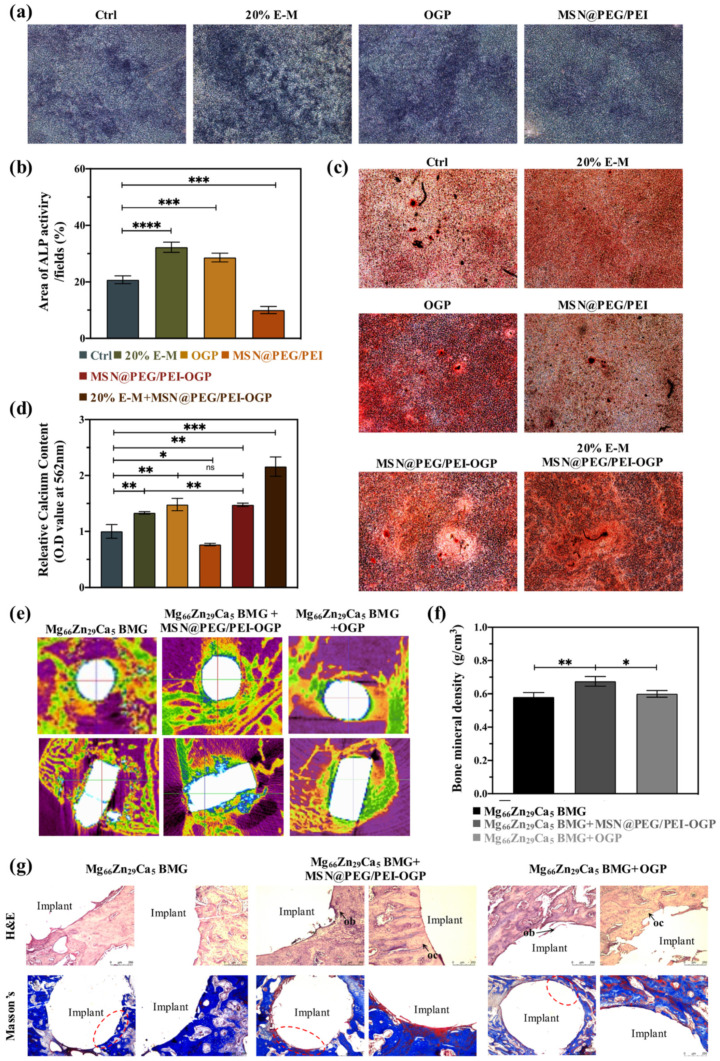
(**a**) Alkaline phosphatase (ALP) activity of induced−MC3T3-E1 cells under different conditions after 14 days of culture. (**b**) Quantitative assay for ALP activity. (**c**) Alizarin red s (ARS) staining of induced-MC3T3−E1 cells after 21 days of culture. (**d**) Quantitative assay for ARS staining. (**e**) Micro−CT image of a distal rabbit femur at 4 weeks post−surgery. (**f**) Quantitative assay for the bone mineral density of distal rabbit femurs surrounding the implant site at 4 weeks post−surgery. (**g**) H&E staining and Masson’s trichrome staining of tissues surrounding the implant site. ob, osteoblasts; oc, osteocytes. Type I collagen was stained blue and mature bone was stained red. The concentration of nanoparticles was 100 μg/mL. (*n* = 3 per group). (Data are the mean ± standard deviation, * *p* < 0.05, ** *p* < 0.01, *** *p* < 0.001, **** *p* < 0.0001).

**Table 1 pharmaceutics-14-01078-t001:** Chemical composition of Mg_66_Zn_29_Ca_5_ bulk metallic glass and metal ions measurements in αMEM and extracted-αMEM (mean ± standard deviation).

		Mg	Zn	Ca
**Molecule Weight**		24.305	65.38	40.078
**EDS (at.%)**	Design	66	29	5
	Result	65.81 ± 0.44	28.69 ± 0.35	5.50 ± 0.34
**ICP-MS (ppm)**	αMEM	0.79	0.02	0.33
	100% extracted-αMEM	60.26	8.08	7.05
	Released metal ions	59.47	8.06	6.72
**The ratio of released metal ions at.%**		89.37	4.50	6.12

## Data Availability

The data presented in this study are available in this article.

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
