# Peer review of "Combining Mg–Zn–Ca Bulk Metallic Glass with a Mesoporous Silica Nanocomposite for Bone Tissue Engineering"

_pharmaceutics, 2022, doi:10.3390/pharmaceutics14051078_

Round 1

Reviewer 1 Report

Overall, the present work is a nice work, but some concerns need to be addressed before accepting the manuscript: 

  1. How does this work go beyond the state-of-the-art? Please highlight it in the introduction. 
  2. What molecular weight has been used for the PEI?
  3. Section 2.2: Please add more details (quantities, temperature, time...). As it is, it is impossible to reproduce. 
  4. Section 2.5: Same as Q3.
  5. Section 2.5 also: If the peptide already has cysteine, why the traut step?
  6. Section 3.2: Authors need to provide characterization of pristine MSNs. Otherwise it is not clear if the different functionalization steps have been successful. Ideally, characterization of each step should be shown (MSNs vs MSN-PEI vs MSN-PEI-PEG vs MSN-PEI-PEG-Peptide). Also, Considering the small molecular weight of PEG-silane, and that PEI's MW is not mentioned, it is not clear to me that if such PEG is doing something. That's why more characterization is needed. 
  7. Authors talk about pore-expanded nanoparticles. However, it seems to me that "pore-expanded" is not the most suitable adjective for this nanoparticles, specially considering that around 3 nm is quite frequent with mesoporous silica. There are several examples of pore-expanded MSNs (pore size >5 nm), see this one for instance: https://doi.org/10.1021/acsnano.6b02819.
  8. Section 3.3: Incubate a material in different conditions is not going to "elevate" the biocompatibility. Please rephrase. 

Reviewer 2 Report

The manuscript presents interesting results concerning degradable metal alloy scaffolds, functionalized with drug loaded mesopourous silica nanocomposite, for bone tissue engineering.

The authors independently deeply investigated degradable metal alloy scaffolds and drug loaded mesopourous silica nanocomposite. However, judging from published literature, it seems that this is the first time that degradable metal alloy scaffolds are combined with mesopourous silica nanocomposite. Therefore, the material and strategy for bone tissue engineering is novel and advanced.

The research has been properly conceived, a wide set of in vitro chemical-physical and biological investigations have been carried out, moreover in vivo animal experiments have also been considered.

In my opinion, the manuscript is interesting for a reader of Pharmaceutics.

Round 2

Reviewer 1 Report

Q1: I am afraid that introducing that sentence does not provide any information on the state-of-the-art and how this new material goes beyond it. 

Q2: OK

Q3: Please indicate the protocols. Researchers should not need to go somewhere else to replicate your results. 

Q4: OK

Q5: If that is the reason, it does not have any scientific rationale, but if the authors have done so, ok. 

Q6: 

I am afraid this is not a valid response. First of all, two of the articles that the authors provide are reviews. In addition, the fact that some nanoparticles have been already reported does not imply that one does not have to characterize them again for a new research project. If that was the case, no one would characterize anything but the final material.

On the other hand, showing a picture with some precipitated nanoparticles does not say anything about the material. Besides, regardless of the stability, the final material needs to be compared with something. Otherwise, as I said before, it is impossible to know if something has happened after the functionalization steps.

Regarding the characterization, normally a N2 sorption analysis can be done with less than 100 mg and, in any case, the sample can be recycled because of the very nature of the technique (introduction and extraction of N2).

I still think the same, the work needs for characterization.

Q7: Even if the molecular weight is now mentioned, I say the same as Q6, the work needs more characterization to assure that what the authors say is actually happening. 

Q8: OK

Q9: OK
